# Targeted Stimuli-Responsive Mesoporous Silica Nanoparticles for Bacterial Infection Treatment

**DOI:** 10.3390/ijms21228605

**Published:** 2020-11-15

**Authors:** Montserrat Colilla, María Vallet-Regí

**Affiliations:** 1Departamento de Química en Ciencias Farmacéuticas, Unidad de Química Inorgánica y Bioinorgánica, Universidad Complutense de Madrid, Instituto de Investigación Sanitaria Hospital 12 de Octubre i+12, Plaza Ramón y Cajal s/n, 28040 Madrid, Spain; 2CIBER de Bioingeniería, Biomateriales y Nanomedicina, CIBER-BBN, 28040 Madrid, Spain

**Keywords:** mesoporous silica nanoparticles, bacterial infection, bacterial resistance, biofilm, targeting, antimicrobials, stimuli-responsive drug delivery

## Abstract

The rise of antibiotic resistance and the growing number of biofilm-related infections make bacterial infections a serious threat for global human health. Nanomedicine has entered into this scenario by bringing new alternatives to design and develop effective antimicrobial nanoweapons to fight against bacterial infection. Among them, mesoporous silica nanoparticles (MSNs) exhibit unique characteristics that make them ideal nanocarriers to load, protect and transport antimicrobial cargoes to the target bacteria and/or biofilm, and release them in response to certain stimuli. The combination of infection-targeting and stimuli-responsive drug delivery capabilities aims to increase the specificity and efficacy of antimicrobial treatment and prevent undesirable side effects, becoming a ground-breaking alternative to conventional antibiotic treatments. This review focuses on the scientific advances developed to date in MSNs for infection-targeted stimuli-responsive antimicrobials delivery. The targeting strategies for specific recognition of bacteria are detailed. Moreover, the possibility of incorporating anti-biofilm agents with MSNs aimed at promoting biofilm penetrability is overviewed. Finally, a comprehensive description of the different scientific approaches for the design and development of smart MSNs able to release the antimicrobial payloads at the infection site in response to internal or external stimuli is provided.

## 1. Introduction

Antimicrobial resistance (AMR) is a major concern threatening human global health [1]. Currently, drug-resistant diseases cause at least 700,000 deaths each year, and this figure could grow to 10 million by 2050 [2]. In fact, it has been foreseen that by this date, more people will die from AMR than cancer [3]. The emergence and expansion of resistant pathogens have been mainly provoked by the extensive misuse of antibiotics both in agriculture and healthcare. A key factor related to the rapid and uncontrolled growth of bacteria is their ability and trend to create biofilms, which consist of communities of microorganisms that grow adhered to a surface and are that are coated by self-produced protective extracellular matrix [4,5,6,7]. Biofilm-forming bacteria exhibit special characteristics that are remarkably different than those of “free-swimming” or planktonic cells, which are responsible of their increased resistance to antimicrobial agents and the immune system [7,8]. Taking into account that the majority of chronic infections (60–80%) are associated with biofilms, the combination of AMR and biofilm formation is a serious clinical concern [9]. An aggravating issue is the lack of new classes of antibiotics in the pipeline, in part, due to the high cost of the development and licensure of antibiotics, and also owing to the complexity involving clinical trials with antibiotics [10]. Thus, the current treatments of infectious diseases are mainly based in the use of the antibiotic classes discovered until the early 1980s [11]. With the aim of increasing the effectivity of the available antibiotics, an increase in either the dose or dosage frequency is required. Unfortunately, this remedy not only increases the toxicity and side effects of the antibiotics, but also promotes the development and spread of bacterial resistance. Therefore, the development of new antimicrobial agents able to efficiently combat bacterial infection and circumvent the abovementioned drawbacks becomes of foremost relevance.

Nanotechnology has entered into this scenario bringing up the opportunity to use nanoparticles (NPs) as efficient nanocarriers for the delivery of antimicrobials [12,13,14,15]. Tackling bacterial infection with nanocarriers provides new opportunities to bypass issues associated with mechanisms of antimicrobial resistance, such as increased drug efflux due to overexpression of efflux pumps [16,17]. Moreover, NPs provide the opportunity to accomplish targeted and stimuli-responsive delivery of antimicrobials, which increases the efficacy of the nanotherapeutics and reduces the off-target toxicity. Inorganic NPs exhibit noticeable advantages compared to their organic counterparts, such as high mechanical strength and good chemical stability under physiological conditions. Moreover, they exhibit excellent biocompatibility, although low degradation rates [18,19]. 

Among inorganic NPs, mesoporous silica nanoparticles (MSNs) constitute one of the most promising nanocarriers for drug delivery, owing their unique characteristics, including large loading capacity, biocompatibility, ease of production and tunable pore diameters and volumes [20,21,22,23,24,25,26]. Moreover, MSNs can be produced in a large scale with variated sizes, morphologies and surface functionalities by using different strategies already described in the literature [27,28,29,30]. These paramount features make MSNs excellent nanoplatforms to assemble different multifunctionalities for the treatment of bacterial infection, highlighting targeted and stimuli-responsive delivery of antimicrobials (Figure 1) [14,31,32,33,34,35,36]. Hence, the attachment of targeting agents to the external surface of MSNs allows specific delivery of antimicrobials to pathogenic bacteria and/or biofilms. Antimicrobial drugs can be incorporated into the mesopores by adsorption from solutions or by grafting some prodrugs to the functional groups of the inner surface of MSNs. Antibiofilm agents can be either adsorbed into the mesopores or linked to the outermost surface of MSNs. Cargo leakage can be hindered by blocking the mesopore outlets gatekeepers able to respond to either internal (bacteria, pH, redox potential, etc.) or external (light, magnetic fields, temperature, etc.) stimuli for on-demand smart drug delivery. Antimicrobial metal NPs can be either embedded in the mesoporous structure or decorate the outermost surface of MSNs. In the case of metal noble NPs (mainly from gold and silver), their well-known antimicrobial action can be reinforced, when needed, by the antibacterial action based on the photo-thermal effect upon proper laser excitation [37,38]. Antimicrobial metal cations can be complexed to ligands grafted to the surface of MSNs [39]. Biocompatible polymers, such as polyethylene glycol (PEG) can be grafted to MSNs to provide them of “stealth” properties. External functionalization with different organic groups can be carried out to modulate surface charge of MSNs and therefore their interaction with bacteria/biofilm. Magnetic NPs for magnetic guidance, triggered cargo release, sensing and biofilm disruption can be also incorporated. Last, grafting of fluorescent molecules can be accomplished for imaging purposes. 

Herein, the scientific advances developed to date in MSNs for infection-targeted stimuli-responsive antimicrobials delivery are reviewed. This review describes the targeting strategies for specific recognition of bacteria. Moreover, the possibility of incorporating anti-biofilm agents to MSNs aimed at promoting biofilm penetrability is tackled. Finally, a comprehensive and detailed description of the different scientific approaches for the design and development of smart MSNs able to release the antimicrobial payloads at the infection site in response to internal or external stimuli is provided.

## 2. Targeted Delivery of Antimicrobials 

### 2.1. Targeting Bacteria

When engineering NPs as antimicrobial delivery systems against bacterial infection, a major challenge is to release the antimicrobial cargo solely at the target site without affecting healthy cells. The goal is to increase the selectivity and efficacy of antimicrobials, reduce antibiotic doses and frequency of the treatment and prevent undesirable side effects associated with unspecific drug delivery. This can be accomplished by either passive or active targeting, or by a combination of both. Passive targeting, which relies on the enhanced permeability and retention (EPR) effect, is a major breakthrough in tumor-targeted nanomedicines for cancer therapy [40,41]. However, although there is also evidence of EPR effect in bacterial infection [42,43], passive targeting of NPs-based antimicrobial systems has been scarcely exploited. On this regard, active targeting becomes a powerful complementary or alternative strategy to provide nanosystems of specificity to the site of infection. It is achieved by decorating the outermost surface of NPs with targeting ligands that produce selective accumulation to targeted sites and therefore differentiate bacteria and healthy cells. This is of foremost relevance in the case of intracellular infections, where bacteria overcome the host immune system by surviving in human cells. With this goal in mind, different approaches have focused on decorating the outermost surface of MSNs with diverse targeting moieties that specifically recognize bacteria but not human host. At this point, it is essential to consider the main difference between bacterial and human cells, i.e., the presence of a cell wall in the former. The bacterial cell wall, which plays an essential role in cell growth, is a protective layer mainly consisting of peptidoglycan and other glycolipids exclusive of bacteria. These distinctive elements become excellent targets in bacteria and can be used to increase specificity and efficiency of bactericidal NPs-based systems [44]. In addition, it is also possible to distinguish between Gram-positive and Gram-negative bacteria attending to the different structure of their bacterial cell wall. The cell wall of Gram-positive bacteria consists in a cytoplasmic membrane covered by a rigid and thick layer of peptidoglycans containing carbohydrate polymers cross-linked via peptide residues [45]. On the other hand, Gram-negative bacteria are surrounded by a triple protective layer, consisting of a cytoplasmic membrane, a thinner and more rigid peptidoglycan layer with shorter cross-links, which is further covered by a hydrophobic lipid bilayer consisting of lipopolysaccharides (LPS). This lipid layer is responsible of the great resistance of Gram-negative bacteria to numerous antimicrobial agents [46]. Thus, by choosing the appropriate targeting moiety, it is feasible not only to discriminate between bacteria and human host cells, but also to guide the targeted nanosystem to a particular type of bacteria. The main targeting strategies reported up to date, the drug loaded and developed MSNs-based nanocarriers, together with the targeted bacteria are collected in Table 1.

Several studies have evidenced that the presence of positive charges on the surface of NPs favors internalization in both Gram-positive and Gram-negative bacteria [59]. The presence of negative charges in the outer bacterial membrane, able to interact throughout electrostatic attractive interactions with positively charged NPs, would explain this fact [60]. In this context, Vallet-Regí and co-workers developed a novel “nanoantibiotic”, based on MSNs decorated with polycationic dendrimers, which was able to penetrate Gram-negative bacteria [47]. To this end, poly(propylene imine) dendrimer of third generation (G3) was covalently grafted to the outermost surface of MSNs, whereas the antibiotic levofloxacin (LVX) was loaded into the mesoporous channels (Figure 2). The high density of positive charges and flexibility on the surface of G3-MSNs produced electrostatic interactions with the negatively-charged *E. coli* bacterial walls, which triggers their permeabilization and thus favors internalization. Moreover, loaded LVX was released in a sustainer fashion at effective bactericidal dosages. These studies demonstrated that the combination of the cell wall disruption capability of G3 dendrimer and the antibiotic effect of LVX into a unique MSNs-based nanosystem has a synergistic antimicrobial effect. Comparable results have been found for Gram-positive *Listeria monocytogene* by using polyamine-functionalized MSNs, as reported by Barat and co-workers [61]. The attractive forces between positive amine corona on MSNs and negative charges in bacteria membrane prompt cell membrane disruption. The high concentration of positive charges from immobilized-amines on the surface of MSNs increased 100-fold their antimicrobial power compared to free polyamines. It is also possible to go a step beyond, by using cationic polymers not only as targeting ligands that increase antimicrobial effect of loaded cargo, but also as responsive pore gatekeepers that hinder premature drug release until exposure to target stimulus. In this line, Martínez-Máñez and co-workers reported the enhancement of the efficacy and broadening the spectrum of antibacterial agents, namely vancomycin (VAN) and histidine kinase autophosphorylation inhibitors (HKAIs), [48,49] by developing ε-poly-L-lysine (ε-pLys)-capped MSNs as bacteria-responsive nanocarriers, as will be discussed in detail in Section 3.1.3. The results indicated that the enhancement of antimicrobials toxicity to Gram-negative bacteria is due to the bacterial wall damage induced by positively-charged ε-pLys, which allows the entrapped cargo to gain access into the cell (vide infra).

In the search of higher specific nanosystems different approaches have exploited the “ligand-receptor binding” concept and decorate the outermost surface of MSNs with ligands that specifically bind surface molecules or receptors overexpressed in bacteria cell wall. Some of the ligands reported in the literature to provide MSNs of targeting ability include antibodies, aptamers, peptides, carbohydrates and small molecules, such as amino acids, vitamins and certain antibiotics.

#### 2.1.1. Antibodies

Antibodies are highly ligands that specifically bind with high affinity to antigens present in the target bacterial cell surfaces [62]. Taking advantage this superior property, Zink and co-workers developed an innovative nanosystem using the FB11 antibody, with high affinity towards antigens of the LPS of pathogenic *Francisella tularensis* (*Ft*), for the treatment of lethal pneumonic tularemia [50]. In this nanosystem, the FB11 antibody played a dual role, as a targeting ligand and as a responsive capping agent. This approach had an added value by avoiding the use of redundant systems and, even more importantly, the need of additional and sometimes complex capping procedures that may compromise the security and biocompatibility of the final nanodevice. This nanosystem, whose description and operating mechanism are detailed in Section 3.1.2, showed promising in vitro and in vivo results for the selective treatment of *Ft* infection. Very recently, antibody-targeted MSNs-based nanosystems have been used in the design of a theranostic platform, integrating diagnosis and treatment elements, for highly-sensitive rapid and accurate bacteria detection and eradication in the bloodstream [51]. To this end, sulfonated-hyaluronic acid (S-HA) terminated magnetic (Fe_3_O_4_ NPs) MSNs (MMSNs), loaded with VAN and decorated with an anti *S. aureus* antibody (Anti-*S. aureus* Ab) by amidation reaction, affording Ab@S-HA@MMSNs. Then, the Ab@S-HA@MMSNs nanosystem was immobilized into a magnetic glassy carbon electrode (MGCE) by magnetic interaction. The specific antigen-antibody binding between *S. aureus* in solution and Anti-*S. aureus* Ab on the surface of MGCE, provoked changes in the electrochemical signals that permitted to accurately determine the amount of *S. aureus* in solution. Moreover, such specificity provided this immunosensor of high selectivity towards *S. aureus*. Thus, when *E. coli* and *P. aeruginosa* were detected, no significant changes of peak current values among the different concentrations were observed, whereas the peak current values of *S. aureus* changed remarkably as the concentrations changed. Moreover, the system exhibited antiadhesion properties due to the anticoagulant property of S-HA terminated MMSNs, which allowed the direct detection and quantification of *S. aureus* in whole blood. Good electrochemical response towards *S. aureus* were obtained in the 10–10^10^ CFU/mL range, with a detection limit of three colony forming units (CFU) per mL and high selectivity, stability and reproducibility. Moreover, this theranostic nanoplatform was efficient as eradicating *S. aureus*, relying on the enzyme-responsive antibiotic delivery capability of the nanoplatform, as will be discussed later.

#### 2.1.2. Aptamers

Aptamers are single-stranded short oligonucleotide sequences that can bind with high affinity to specific targets, namely proteins, peptides, carbohydrates, small molecules, toxins or even live cells, and exhibit great potential in the detection and treatment of microbial infections [63]. Aptamers fold via intramolecular interactions, creating tertiary conformations that specifically recognize diverse types of targets. They are powerful alternatives in the design of targeted nanotherapeutics owing to their small size, non-immunogenicity, ease to be synthesized, characterized and modified and high specificity and affinity towards their target similarly to antibodies. On this line, Ozalp and co-workers proposed an innovative nanosystem based in aptamer-modified MSNs able to selectively target and eradicate *S. aureus* bacteria. To this end, MSNs were loaded with VAN and decorated with the SA20hp aptamer, which exhibits high affinity towards the surface antigens present in *S. aureus* [52]. The nanosystem was able to selectively recognize the target bacteria, releasing the antibiotic cargo throughout an antigen-triggered mechanism, as detailed in Section 3.1.2. The dual role played by SA20hp, i.e., targeting ligand and antigen-responsive nanocap, allowed for developing in a simple manner nanodevices with high selectivity and antimicrobial efficacy towards the target bacteria. 

#### 2.1.3. Peptides

Peptides, consisting of linear or cyclic sequences of less than 50 amino acids, show several benefits compared to proteins, their larger counterparts, including easier synthesis and conjugation and also higher stability and resistance to the environment. On the other hand, peptides are much more inexpensive and easy to manufacture than antibodies. In addition, peptides are less prompt to elicit immune response by the host than antibodies. All these characteristics make peptides excellent choices to be incorporated as active targeting ligands to NPs [64]. In this respect, Tang and co-workers devised an active targeting strategy involving the cationic human antimicrobial peptide fragment ubiquidin (UBI)_29–41_, which exhibited high specificity towards *S. aureus* [53]. MSNs were loaded with gentamicin (GEN), capped with a lipid layer (as the microenvironment sensitive blocking cap) and decorated with (UBI)_29–41_, affording GEN@MSN-LU nanosystems. In vitro experiments carried out both in planktonic bacteria and in *S. aureus*-infected preosteoblasts (MC3T3-E1) and macrophages (RAW 264.7) showed the excellent targeting and antimicrobial effectiveness of the nanosystem. Notably, liposome and UBI_29–41_ modification of MSNs improved internalization into mammalian cells. Thus, once internalized, the NPs migrate to the infection site within the phagocytic cells and there, the presence of certain bacterial toxins triggers GEN release producing pathogenic bacteria death. Moreover, a downregulation of the inflammation-related gene expression in infected preosteoblasts or macrophages after GEN@MSN-LU NPs administration is an added value to eliminate inflammation produced by intracellular bacterial infection. Finally, the excellent results achieved in vivo in mouse models supported this peptide-targeting strategy as a promising alternative to treat intracellular infections. A similar approach was reported by Jayawardena and co-workers, who developed antimicrobial peptide (LL-37)-targeted MSNs as colistin (COL) delivery systems against extracellular and intracellular *Pseudomonas aeruginosa* infections [54]. To prevent premature cargo release, COL-loaded MSNs were coated by a liposomal shell and subsequently conjugated with a *P. aeruginosa* peptide LL-37, affording COL-MSN@LL-(LL-37). The capability of LL-37 to recognize the outer membrane of *P. aeruginosa* allowed the nanosystem targeting intracellular bacteria without eliciting cytotoxic effect on the mammalian cells. Notably, the antimicrobial efficacy of COL encapsulated in targeted nanosystem was 6.7-fold higher than that of free COL, being only 7% the bacterial viability after treating lung epithelial cells with COL@MSN@LL-(LL-37). 

#### 2.1.4. Carbohydrates

Carbohydrates are interesting targeting ligands owing to their involvement in different selective binding and transport processes in bacteria. Among carbohydrates, trehalose (Tre) has received the attention of different research groups because it plays an important role in bacteria of the genus *Mycobacterium*, including the tuberculosis (TB) pathogen [65]. Tre is abundant as free form in the cytosol of *Mycobacteria* and as glycolipids in the cell wall. Selective trehalose uptake have been observed in *Mycobacteria*, making this saccharide good alternative for targeting purposes [66]. Ramström and co-workers designed Tre-targeted antimicrobial MSNs for selective targeting and killing of *Mycobacteria smegmatis* [55]. MSNs were functionalized with perfluorophenilazide silane (PFPA-Si) and then decorated with Tre via azide-mediated surface photoligation, affording M-PFPA-Tre. Finally, the NPs were loaded with isoniazid (INH), an antibiotic widely used for the treatment of TB, obtaining M-PFPA-Tre-INH. The nanosystem exhibited increased bacteria-killing capability and the advantage of displaying the target function of trehalose to promote the localized release of INH. Thus, the M-PFPA-Tre-INH nanosystem produced a complete bacterial growth inhibition at concentrations in the 3–4 mg/mL range, whereas higher concentrations were needed (4.5–5 mg/mL) when using a preparation resulting from the direct mix of MSNs and INH. Yan and co-workers used the same strategy to graft Tre, as the targeting ligand, to hollow oblate MSNs (HOMSNPs), which consisted of NPs with hollow interior and permeable mesoporous shell [41]. Upon INH loading, the antimicrobial effect and the targeting selectivity of the resulting nanosystem (HOMSNs-Tre-INH) were evaluated in vitro. This nanosystem exhibited enhanced antibacterial activity against *M. smegmatis*, which was attributed to the increased interactions between NPs and mycobacteria as a result from the targeting effect of Tre. On the other hand, the targeting selectivity of HOMSNs-Tre-INH was tested by means of two different experiments: the first one consisted in replacing Tre by mannose as a control against *M. smegmatis*; the second one involved the exposition of Tre-targeted NPs to two different strains of Gram-negative (*E. coli*) and Gram-positive (*S. epidermidis*). Both internalization and bacteria cell death were much lower, or even non-existent, than those observed in *M. smegmatis*. 

#### 2.1.5. Small Molecules

The preparation of small molecules exhibiting different structures and characteristics is relatively inexpensive, making them attractive targeting ligands. On this regard, Raichur and co-workers developed arginine (Arg)-functionalized nanosystems for targeting and treating intracellular *Salmonella* [56]. The nanosystem was synthesized using a layer-by-layer (LBL) coating approach. First, the negatively charged MSNs were coated with the positively charged cationic polymer protamine and then with the negatively charged pectin polyelectrolyte. Finally, L-Arg was conjugated to the exterior pectin and the resulting NPs were loaded with the antibiotic fluoroquinolone ciprofloxacin (CIP) to obtain CIP-Arg-MSNs. Gradual CIP release over a period of 24 h from CIP-Arg-MSNs was observed. In vitro assays in *Salmonella* infected macrophages revealed two-fold higher antibacterial activity with CIP-Arg-MSNs compared to free CIP. The co-localization of NPs with the intravacuolar *Salmonella* and localized antibiotic delivery would explain the increased antibacterial activity of the developed NPs. In vivo bacterial burden and morbidity studies in

BALB/c mice showed nearly tenfold lower *Salmonella* burden in infected organs, such as spleen, liver, and mesenteric lymph nodes. Comparable survival rates were observed at a lower CIP-Arg-MSNs dosage over free CIP.

Taking advantage of the overexpression nutrients receptors in bacterial surface it is possible to decorate NPs with vitamins for targeting purposes. Using this approach, Liu and co-workers prepared pH-responsive antimicrobial nanosystems functionalized with folic acid (FA) as the targeting ligand [57]. FA was chosen as targeting ligand due to its affinity towards folate receptors (FRα), which are overexpressed in the surface of bacteria [67]. The nanosystem consisted in MSNs loaded with ampicillin (AMP), capped with a calcium phosphate (CaP) layer for pH-responsive release, and covered with double FA layer, (MSN@FA@CaP@FA), as described in Section 3.1.5. In vitro experiments showed that the nanosystem effectively increased the uptake and reduced the drug efflux effect in *E coli* and *S. aureus* due to the specific targeting of FA. In vivo experiments in infected mice showed that MSN@FA@CaP@FA efficiently reduced the mortality of AMP-resistant *E. coli* infection, and promoted wound healing of AMP-resistant *S. aureus* infection.

In another study by Wang et al., the antibiotic VAN was grafted to MSNs to develop nanosystems able to selectively target and eliminate Gram-positive bacteria over macrophage-like cells [58]. The skeleton of VAN strongly binds, throughout hydrogen bond interactions, with the terminal D-alanyl-D-alanine moieties of Gram-positive bacteria, which was used for selective targeting purpose. In addition, VAN inhibits bacterial growth by hindering the normal development of the cell wall, which was used for bactericidal purpose. Thus, MSNs were functionalized with aminopropyl groups and then VAN was grafted via amidation reaction. In vitro experiments showed that the nanosystem selectively recognized Gram-positive *S. aureus* in the presence of Gram-negative *E. coli*. On the other hand, the nanosystem produced a significant inhibition of *S. aureus* growth without decreasing cell viability of macrophage-like cells. Lastly, the antibacterial capability of the nanosystem was evaluated in vivo in a mice model revealed a remarkable decrease of bacteria in *S. aureus* infected tissues.

### 2.2. Targeting Biofilm

We have reviewed the different targeting strategies reported to date aimed at increasing the specificity and efficacy of antimicrobial delivery from MSNs against bacterial infection involving planktonic bacteria, i.e., isolated bacteria floating in solution. However, the challenge we face is different when the same kind of bacteria are associated in communities forming biofilms. Bacteria in biofilms generally develop resistance to antimicrobial agents, since they are embedded in a self-produced protective matrix of extracellular polymer substances (EPS), composed of extracellular DNA, polysaccharides, proteins, glycolipids and other ionic molecules [4]. Targeting bacterial biofilms with MSNs able to disrupt the EPS, penetrate bacterial biofilm and release the antimicrobial cargo, constitutes a promising alternative to eradicate bacterial biofilms. This is an emerging research field and there are still few publications on this topic. Table 2 summarizes biofilm-targeted MSNs as antimicrobial delivery systems. 

Several studies have focused on engineering MSNs for the delivery of antibiofilm agents able to reduce EPS cohesiveness and disperse the biofilm biomass, such as certain proteins including lysozyme [70] or DNase I [71]. An alternative strategy consists on tailoring the nanoparticle-biofilm interactions by taking into account that EPS components exhibit (typically) negatively charges [72]. In this line, the group of Prof. Vallet-Regí reported the design of “nanoantibiotics” consisting of antibiotic-loaded MSNs functionalized with positively charged moieties as the targeting agents [47,68]. Thus, MSNs were externally modified with N-(2-aminoethyl)-3-aminopropyltrimethoxy-silane (DAMO) and loaded with the antibiotic LVX [68]. Amine functionalization provided MSNs of positive charges that improved the ability of the nanosystem to target and penetrate *S. aureus* biofilm. In addition, microbiological studies showed complete biofilm destruction when antibiotic and targeting agent are synergistically combined in a unique nanoplatform. On the other hand, high anti-biofilm efficiency against Gram-negative *E. coli* bacteria was also accomplished throughout the synergistic combination of polycationic dendrimers (G3), as bacterial membrane permeabilization agents, and LVX-loaded MSNs [47]. 

The same research group proposed an alternative biofilm-targeting strategy consisting of decorating the outermost surface of MSNs with molecules showing affinity towards certain components present in the EPS [69]. In this case, the lectin Concanavalin A (ConA) was chosen as the targeting ligand, owing to its capability to recognize and bind to glycan-type polysaccharides present in the biofilm EPS. MSNs were functionalized with carboxylic acid groups, decorated with ConA and loaded with LVX. The bacterial biofilm-targeting efficacy of the nanocarrier was evaluated in *E. coli* biofilms. A dose dependent internalization was observed, i.e., the greater the concentration of added NPs, the higher the amount of NPs able to penetrate the bacterial biofilm. Moreover, the synergistic combination of ConA and LVX in the same nanoplatform lead to a complete biofilm destruction. In this regard, ConA-driven penetration of the nanosystem into the biofilm allows for the release of the antibiotic inside, producing high antimicrobial efficacy (Figure 3).

## 3. Stimuli-Responsive Antimicrobials Delivery 

Although the advantageous features of MSNs as drug delivery systems are numerous, their porous structure does not offer controlled release of the entrapped antimicrobial cargoes. Zero premature release of the hosted drugs can be accomplished by incorporating different organic or inorganic moieties into the pore openings, behaving as gatekeepers or nanocaps. The stimuli-responsive or “smart” performance can be accomplished by using cleavable-linkages to graft these moieties or by using gatekeepers that undergo chemical and/or physical changes in response to given stimuli. Thus, smart nanosystems would load, protect and transport the antimicrobial to the target site where, in the presence of an internal or external stimulus, release the the antimicrobial payload. Stimuli-responsive MSNs have been widely proposed for antitumor therapy [24,73,74,75]. Nonetheless, the research on smart MSNs for the treatment of bacterial infections started a few years ago and, therefore, the number of reports into this area is not so extensive. The different stimuli that have been exploited to trigger antimicrobials delivery from MSNs can be classified into internal (presence of bacteria, bacterial toxins, pH, redox potential and dual stimuli) or external (chemical species, temperature, light and alternating magnetic field). In this section we overview the most relevant studies involving stimuli-responsive MSNs for bacterial infection treatment focusing on the triggering stimulus depending on the specific characteristics of the infection to treat. 

### 3.1. Internal Stimuli-Responsive MSNs

#### 3.1.1. Presence of Bacteria

The first report on MSNs as stimuli-responsive delivery systems of antimicrobials was authored by Martínez-Máñez and co-workers, who developed MSNs able to release the cargo in the presence of bacteria [48]. In this study, MCM-41-type MSNs were loaded with VAN, and their outer surface was functionalized with N-[(3-trimethoxysilyl)propyl] ethylendiamine triacetic acid trisodium salt (TMS-EDTA). Finally, the mesopores were capped with the cationic polymer ε-pLys via electrostatic interactions with the negatively charged NPs surface. The antimicrobial effect of this nanodevice was tested against different Gram-negative bacteria, namely, *E. coli*, *Salmonella typhi* and *Erwinia carotovora.* As is schematically shown in Figure 4, the presence of bacteria triggers pore uncapping, due to the adhesion of the ε-pLys gatekeeper with the negatively charged bacteria wall, which allows the release of the entrapped cargo. One of the major outcomes of this pioneering approach is the synergistic enhancement of antimicrobial effect, in terms of growth inhibition and the cell viability of both ε-pLys and VAN, which does not occur with the free drugs. The other remarkable achievement is the broadening of the antibacterial spectrum of action of VAN, whose solely administration triggers bacterial resistance in Gram-negative bacteria. Authors postulate that the improvement of VAN toxicity to Gram-negative bacteria when this nanoformulation is used can be ascribed to the bacterial wall damage provoked by ε-pLys adhesion, which allows the loaded VAN to penetrate into the cell.

Later on, the same research group used a similar approach to load histidine kinase autophosphorylation inhibitors (HKAIs) into stimuli-responsive MSNs [49]. HKAIs-loaded MSNs were functionalized with TMS-EDTA and capped with ε-pLys. The antimicrobial activity of this nanosystem against *E. coli* and *Serratia marcescens* Gram-negative bacteria was higher than that of free HKAIs. As described above, the operating mechanism of this nanodevice is based on the interaction of the positively charged capped MSNs with the negatively-charged bacterial cell wall, which triggers pore uncapping and cargo release and consequently inhibits bacteria growth. This nanosystem did not exhibit adverse effects on mammalian cells viability or immune function of macrophages and did not show any toxicity to zebrafish embryos in vivo. 

More recently, Khashab and co-workers developed a smart mixed-matrix membrane coating for X-ray dental imaging devices with the capability of detecting and inhibiting healthcare-associated infections [76]. This antibacterial coating was formed by a poly(ethylene oxide)/poly(butylene terephthalate) polymer matrix containing homogeneously-dispersed antimicrobial nanofillers. These nanofillers were composed of aminopropyl-modified MSNs (MSN-NH_2_) loaded with kanamycin (KANA) antibacterial agent and capped with LYS-functionalized gold nanoclusters (AuNC@LYS). Bacteria-responsive nanofillers were obtained by the self-assembly of positively charged MSN-NH_2_ and negatively charged and fluorescent AuNCs. The presence of bacteria triggered AuNC@LYS detachment from MSNs, due to the interaction of LYS with the bacterial cell wall, allowing the release of the entrapped KANA antibacterial agent. Simultaneously, variation and eventually disappearance of the red fluorescence of AuNC@LYS under UV light was detected, providing a qualitative bacterial sensing nanoplatform. Moreover, bacteria-triggered KANA release performance of this smart coating was investigated in the presence of *E. coli*. The results revealed that the viability decreased by 50% only after 50 min, and by ca. 80% after 170 min of assay. Effective antibacterial activity towards *Bacillus safensis* was also probed. This smart coating successfully allowed the detection and inhibition of bacterial colonization and growth on a common radiographic dental imaging device (photostimulable phosphor plate), a substrate susceptible to contamination by oral bacteria. This strategy could be expanded to many other medical devices without compromising their function.

One of the constraints of the above-described nanosystems is their lack of specificity, which hampers their use for detection and treating a targeted pathogen. An attractive alternative strategy was proposed by Ozalp and co-workers [52], who designed aptamer-gated MSNs able to release the loaded antibiotic VAN upon interaction with specific antigens present on the surface of *S. aureus* bacteria (see Section 2.1.2). To obtain a *S. aureus* specific molecular nanogate, a SA20hp aptamer was grafted to MSNs and converted to hairpin structure, affording SA20hp. The interaction with *S. aureus* surface antigens produces a disruption rearrangement of the aptamer structure that lead to pore uncapping and allowed the release of the entrapped VAN. The antimicrobial efficacy of the nanodevices was in vitro evaluated using *S. aureus* (targeted) and *S. epidermidis* (non-targeted) bacterial cultures. The minimum inhibitory concentration (MIC) values were determined, being 0.420 µg/mL and 6.295 µg/mL for *S. aureus* and *S. epidermidis*, respectively. These results revealed a 15-fold increased efficacy in *S. aureus* compared to that of *S. epidermidis*, which accounted for the specific targeting and higher toxicity in the former. The MIC values of solely VAN were 1.1 µg/mL and 2.4 µg/mL for *S. aureus* and *S. epidermidis*, respectively. These results demonstrated that *S. epidermidis* is only damaged by relative high doses of VAN-loaded aptamer-gated MSNs (6.295 µg/mL). So, it is possible to use appropriate doses (e.g., 0.420 µg/mL) to efficiently kill the targeted bacteria *S. aureus* without harming non-targeted bacteria (*S. epidermidis*). 

Zink and co-workers also proposed an interesting strategy to design a pathogen-targeted detection and delivery nanoplatform based on highly specific antigen-antibody interactions, as commented in Section 2.1.1 [50]. To this end, the O-antigen of the LPS of *F. tularensis (Ft)* bacteria was immobilized on the surface of fluorescein MSNs, and then, the nanosystem was capped by using the FB11 antibody. The presence of the native antigen *Ft* LPS, which exhibits greater affinity towards the capping antibody, produces the competitive displacement of the antibody cap and allows cargo release from the mesopores. The dye used as model cargo allowed evaluating the performance of the nanosystem by measuring the fluorescence levels after incubation with Ft and *Francisella novocida* (*Fn*). After 1 h of assay, the intensity detected in Ft was 100 a.u. whereas that of *Fn* was only 20 a.u., evidencing the high selectivity of the nanodevice. 

#### 3.1.2. Bacterial Toxins

Some studies have focused on developing MSNs able to release the antibiotic payloads in response to the high levels of certain toxins produced and secreted by bacteria in infected microenvironments [77]. In this line, Gao and co-workers designed biohybrid nanomaterials able to release antimicrobial cargo in the presence of hyaluronidase, an enzyme produced by several pathogenic *S. aureus* [78]. The nanohybrids were synthesized following the LBL self-assembly method. First, amoxicillin (AMO), a broad-spectrum antibiotic, was loaded into MSNs. The resulting NPs were functionalized with carboxylate groups for the subsequent adsorption, via electrostatic interactions, of LYS, a bacteriolytic enzyme toward Gram-type positive strains. Then, the NPs were sequentially coated with hyaluronic acid (HA), as the hyaluronidase responsive layer, and with 1,2-ethanediamine-modified polyglycerol methacrylate (EDA-PGMA), a cationic polymer that does not show specific target toward the bacteria but experiences electrostatic attracting interactions with the negatively-charged bacterial membrane (Figure 5). In vitro assays probed that hyaluronidase lead to the specific cleavage of the nanosystem, followed by the release of LYS and AMO. In vitro antibacterial activity tests were carried out against AMO-resistant *E. coli* and *S. aureus* with several NPs concentrations. Due to the protection of the LPS layer surrounding cell membrane in Gram-negative bacteria, LYS exhibited a relatively weak antimicrobial activity for *E. coli*. Thus, better inhibition effect of the nanosystem was obtained towards S. aureus than that of E. coli. The MIC values toward AMO-resistant bacteria were much lower than that of isodose LYS and AMO. The antimicrobial effect of the nanosystem was also in vivo evaluated in a mouse wound model infected with *S. aureus*. Epidermal antimicrobial delivery with a single dose of LYS and AMO significantly reduced the number of bacteria to a bacteriostatic rate of 67.4%, compared to the negative controls, whereas the application of the nanosystems inhibited almost totally the bacterial growth, with a bacteriostatic rate of 99.9%. Wang and co-workers also applied the HA-capping strategy to prepare theranostic MSNs-based nanoplatforms for the detection and killing of *S. aureus* in the whole blood, as has been detailed in Section 2.1.1 [51]. The nanosystem consisted of magnetic MSNs (nanoplatforms) loaded with VAN (antibiotic cargo), capped with sulfonated-HA (capping agent), and decorated with a *S. aureus* antibody (active targeting ligand). In vitro experiments demonstrated that the nanosystem was able to release the entrapped VAN upon addition of hyaluronidase, due to the degradation of the capping agent HA. Moreover, in the presence of *S. aureus*, an “on-demand” antibiotic release was achieved, which allowed to efficiently eradicate these bacteria from the infected bloodstream.

Another strategy relied on the use of lipases, phosphatases and phospholipases secreted by bacteria, to trigger antibiotic release. To this end, MSNs were coated with a lipid bilayer shell as the sensitive-capping element that prevents premature cargo release, protecting the antibiotic from inactivation and overcoming the cellular barriers when addressing intracellular infections [53,54]. Decorating the outermost surface of the NPs with a specific bacteria-targeting peptide, as discussed in Section 2.1.3, conferred the nanosystems of capability to target the site of bacteria. Once there, the liposomal coating was degraded by toxins and the antibiotic molecules were rapidly released to subsequently kill intracellular bacteria cells. It was observed that a remarkable increase in the antimicrobial efficacy compared to the free antibiotic. 

Very recently, Cai and co-workers reported the fabrication of a glutamyl endonuclease-responsive MSN-based nanoplatform to treat *S. aureus*-associated osteomyelitis infections, meanwhile, promote bone tissue regeneration [79]. Glutamyl endonuclease (V8 enzyme) was chosen as antimicrobial release trigger taking the advantage of the overexpression of this enzyme in the microenvironment of *S. aureus* infection. For the synthesis of the nanosystem Ag NPs were first encapsulated into MSNs. Then, poly-L-glutamic acid (PG) and polyallylamine hydrochloride (PAG) were assembled on MSN-Ag by using the LBL assembly method. In this nanosystem PG, a homogeneous polyamide containing an amide linkage, was the sensitive capping element. In the presence of V8 enzyme secreted by *S. aureus*, PG was degraded and the multilayer film outside MSN-Ag NPs was destroyed, which produced pore uncapping and self-adaptive release of loaded Ag NPs and ions. In vitro antibacterial experiments demonstrated the excellent antibacterial effect of LBL@MSN-Ag NPs deposited onto Ti substrates. In vivo experiments in a bacterium infected femur-defect rat model showed that the modified Ti implants were efficient at treating bacterial infection. Moreover, significant formation of new bone tissue was observed, which could be related to the role played by PG in accelerating bone formation. 

#### 3.1.3. pH

Similarly to some tumor tissues, bacterial infection is accompanied by a local drop in pH through a combination of low oxygen triggered anaerobic fermentation [80] and natural immune response of the host eliciting inflammatory responses [81]. The pH at the site of infection can reach values as low as 5.5, [82], which can be efficiently used as release trigger in pH-responsive drug delivery nanosystems. This fact motivated Lee and co-workers to design pH-triggered MSNs where silver-indole-3 acetic acid hydrazide (IAAH-Ag) complex, used as model drug, was linked to MSNs via pH-cleavable hydrazone bond [83]. Upon exposition to an acidic pH of ca. 5.0, the nanosystem released significant amounts of Ag^+^ (70%), whereas moderate release was found (25%) at the physiological pH of 7.4. Positive antimicrobial effect was obtained in planktonic cells and biofilms of bacteria. Thus, NPs at a concentration of 120 µg/mL totally inhibited bacterial *E. coli* and *S. aureus* growth. In addition, NPs at a concentration as low as 30 µg/mL were enough to inhibit biofilm formation of *E. coli*, *B. subtilis*, *S. aureus* and *S. epidermidis*. Moreover, good antibacterial activity in vivo was assessed in intraperitoneal *E. coli* infected adult mice. 

Qu and co-workers developed a “sense-and-treat” hydrogel for the detection and eradication of bacteria [84]. In this report MSNs acted as scaffolds for a fluorescence probe and an antibacterial drug. Thus, MSNs were loaded with VAN and end externally functionalized with fluorescein isothiocyanate (FITC). Then, a copolymer resulting from the copolymerization between a rhodamine B derivative (RhBAM) and the pH-sensitive polymer poly(N-isopropyl acrylamide-co-acrylic acid) (PNIPAAM) was grafted onto MSNs. The copolymer on MSNs was in an initial swollen-state at the physiological pH, whereas it started to shrink under acidic environment, acting as nanogate to control drug delivery (see Figure 6). In this system, FITC and RhBAM made up the radiometric fluorescence probe. Owing to the pH-sensitive properties of FITC, the nanosystem had a strong emission at 518 nm at neutral or basic pH, while this emission was reduced at acid pH. On the other hand, at neutral or basic pH RhBAM showed no fluorescence, while at acidic pH this moiety emitted a strong fluorescence at 575 nm. Therefore, the acidic pH at the site of infection triggered both a fluorescence change and VAN release to kill bacteria. VAN release studies conducted on NPs with and without RhBAM at different pH showed higher VAN release at acidic pH from RhBAM-containing NPs. On the contrary, RhBAM-free NPs showed similar antibiotic release profiles independently of the tested pH. The successful “sense-and-treat” efficacy of the nanosystem was probed on *E. coli* cultures, wherein NPs concentration of 280 µg/mL showed high antimicrobial activity during 36 h.

Li and co-workers developed a pH-responsive co-delivery system of β-lactam antibiotic (carbenicillin, Car) and a β-lactamase (sulbactam, Sul) to eradicate methicillin-resistant S. aureus (MRSA) [85]. Fe^3+^-Car metal organic framework (CarMOF) acted as capping units when grafted to sul-MS-loaded MSNs. Cargo (Car and Sul) release experiments were carried out at pH 7.4 and 5.0, evidencing the higher drug release at pH 5.0, in good agreement with the acid-triggered dissolution of CarMOF. CarMOF-Sul-MSNs system inhibited MRSA growth at acidic pH owing to the co-delivery of Car (from the degradation of CarMOF) and Sul (loaded into the mesopores). In addition, no cytotoxicity was observed after incubation with RAW 264.7 cells for 48 h. Finally, in vivo assays showed decreased MRSA infection in the skin of mice treated with these NPs.

In another study, Liu and co-workers described the design of a pH-responsive nanosystem consisting of ampicillin (AMP)-loaded MSNs coated with double FA and CaP (see Section 2.1.5) [57]. First, MSNs were modified by grafting FA via electrostatic interactions and then coated with CaP through chelate effect and biomineralization. Finally, another FA layer was grafted to the outermost surface of the NPs. The targeting capability of FA towards the bacterial infection site efficiently increased the NPs uptake and overcame the efflux pump effect in *E. coli* and *S. aureus*. In vitro tests in mammalian cells cultures showed the biocompatibility of this nanosystem. Moreover, no significant hemolytic activity was observed in human blood even at NPs concentrations of up to 160 µg/mL. In vivo assays demonstrated the antibacterial efficacy of this nanosystem, which effectively diminished the mortality of drug-resistant *E. coli* infection and promoted wound healing process in drug-resistant *S. aureus* infection. 

Very recently, Shoueir and co-workers reported the synthesis of MSNs loaded with LVX and coated with polylactic acid nanoflowers (PLA-NF) as biodegradable pH-responsive gatekeeper [44]. The structure of PLA nano-shells highly depended on the pH of the environment. Under acidic pH as low as 2.01 the nano-shell was hydrolyzed and degraded, allowing LVX release out of the open mesopores. At neutral pH of 7.0 PLA nano-shells became insoluble and collapsed onto MSNs surface, creating a compact capping layer that slowed down the LVX release rate. The antimicrobial efficacy of this nanosystem against *S. aureus*, *E. coli* and *Candida albicans* (*C. albicans)* was demonstrated in vitro. Moreover, the cytocompatibility and non-toxicity of this nanosystem was also confirmed in human osteoblast cell lines. 

pH-responsive MSNs have been proposed for the intracellular delivery of antimicrobials to bacteria-infected cells in the treatment of intracellular infections, such as tuberculosis [86,87] or tularemia [88]. Since macrophages are the primary host cells that harbor *intracellular pathogens* [89], the possibility to introduce antibiotics selectively into macrophages would increase the therapeutic efficacy and reduce side effects associated with conventional treatments. It is possible to take advantage of the efficiency of macrophages in internalizing NPs and delivering such NPs to acidified endosomes, to design pH-responsive MSNs for microbial delivery. In this regard, Zink and co-workers developed INH-delivery MSNs equipped with pH-operated beta-cyclodextrin (β-CD) nanovalves fabricated by covalent attachment of molecular threads to the pore mouths of MSNs and addition of bulky β-CD molecules that at neutral pH bonded to the threads blocking the mesopores [86]. At acidic pH, the protonation of the molecular threads takes place, which reduces the binding affinity between β-CD capping molecules and the thread and therefore nanovalves open, which allows INH release within *M. tuberculosis*-infected macrophage. In vitro assays demonstrated that INH delivered from the developed nanosystem kills 1.5 logs more intracellular *M. tuberculosis* than equivalent doses of free INH. In another work, the same research group developed a pH-responsive nanosystem based on MSNs with a pH-sensitive nanovalve, whose operating mechanism is similar to that above described, for the delivery of the fluoroquinolone antibiotic moxifloxacin (MXF) and evaluated its efficacy in the treatment of *F. tularensis* infected macrophages [88]. The nanovalve was closed at the pH of 7.4, trapping the cargo inside the mesopores. Protonation of the organic stalks as the pH decreases to values of 6 and lower, weakened their interaction with capping molecules. The nanosystem showed high efficacy in killing *F. tularensis* in vitro, in cell cultures of infected macrophages, and in vivo, in a mouse model of lethal pneumonic tularemia, with much more effectivity than an equivalent amount of free MXF.

The same research group developed pH-responsive nanosystems carrying covalently grafted INH throughout pH-responsive hydrazone bonds [87]. The nanosystems consisted in MSNs functionalized with aldehyde groups and loaded with INH as a “prodrug”, through the chemical grafting of the hydrazine portion of INH to the aldehyde-modified MSNs, affording a pH-sensitive hydrazone bond. The hydrazone bond is stable at neutral pH, whereas it has a high INH release capability under acid pH, such as that of the endolysosomal compartment of macrophages after NPs uptake. Finally, to increase the dispersibility of the nanosystem and promoting endosomal escape following uptake, INH-loaded MSNs were coated with the copolymer poly(ethylene imine)-poly(ethylene glycol) (PEI-PEG). In vitro evaluation demonstrated that the developed nanosystems were avidly ingested by *M. tuberculosis*-infected human macrophages and killed the intracellular bacteria in a dose-dependent fashion. In addition, it was probed in a mouse model of pulmonary tuberculosis that the NPs were well tolerated and showed greater efficacy than that achieved with equivalent doses of free INH.

#### 3.1.4. Redox Potential 

The reducing power of living cells is higher than that of extracellular medium or plasma due to the great number or redox couples that are kept mainly in the reduced state by diverse metabolic processes. Among these redox couples, glutathione (GSH)/GSSG is the most abundant inside cells [90]. This fact inspired Zink and co-workers to develop redox-responsive antimicrobial nanosystems for the eradication of intracellular bacterial pathogens, concretely *F. tularensis* in macrophages [91]. For this purpose, antibiotic MXF was loaded into MSNs and a redox-sensitive disulfide snap-top acted as the gatekeeper (Figure 7). MSNs were functionalized with mercaptopropyl groups and with adamananethiol forming a disulfide bond. Then NPs were loaded with MXF and capped by addition of β-CD, which formed an inclusion complex with adamantanethiol. The exposition to reducing milieus (e.g., after GSH addition or after macrophages uptake) triggered the cleavage of disulfide bond and cargo release was observed. To evaluate whether these disulfide snap-tops were able to operate at physiological concentrations of GSH, NPs were loaded with Hoechst fluorescent dye. The results indicated that cargo was released intracellularly, staining the nuclei of macrophages due to the presence of GSH inside the cells. The nanosystem was efficient at killing *F. tularensis* in macrophages. Thus, whereas, without treatment, the bacteria grew 2.4 logs in 24 h, redox-responsive MXF-loaded MSNs (6.25–400 ng/mL) or MXF (1–64 ng/mL) reduced bacteria in macrophages in a dose-dependent fashion. MOX delivered by the disulfide snap-top MSNs had similar efficacy than free MXF in the in vitro *F. tularensis*-infected macrophague model. In vivo studies in a mouse model of lethal pneumonic tularemia showed that the nanosystem prevented weight loss, illness and death, noticeably reducing the burden of *F. tularensis* in the lung, liver and spleen and showed higher efficacy than a comparable amount of free antibiotic. 

Among redox stimulus, reactive oxygen species (ROS) is present in living organisms and mainly consists of superoxide (O_2_^−^), hydrogen peroxide (H_2_O_2_), hypochlorite (OCl^−^), peroxynitrite (ONOO^−^) and hydroxyl radical (–OH) [92]. Free oxygen radicals are highly toxic to pathogens and they are used as a powerful tool to prevent colonization of tissues by pathogenic microorganisms [93]. ROS overproduction in the bacterial infection sites is receiving growing attention to attain ROS-responsive delivery nanosystems antimicrobials. In this regard, Chen and co-workers designed a nanosystem consisting of amino-functionalized MSNs loaded with VAN and grafted with thioketal (TK) functionalized methoxy poly(ethyleneglycol) (mPEG-TK) as ROS-responsive gatekeeper (Figure 8). The interaction with the ROS in the microenvironment triggers the cleavage of TK linker and polymeric cap degradation, which produced pore uncapping and allowed release of encapsulated VAN. In vitro antibacterial efficacy tests against *S. aureus* revealed that this nanosystem exhibited better antibacterial activity than free-VAN, due to its robust influence on the disintegration of the bacterial membrane. Moreover, the good in vitro biocompatibility in osteoblast cell cultures and the effective antibacterial ability in the healing of skin wounds in rats exposed to *S. aureus*, probed the suitability of these NPs in the treatment of infected wounds.

#### 3.1.5. Dual Stimuli

Tang and co-workers developed a GSH/pH dual-responsive antimicrobial nanosystem for the co-delivery of chlorhexidine (CHX) and Ag^+^ in response to a pathological environment in the oral cavity [94]. In this study, disulfide-bridged MSNs were decorated with Ag NPs (Ag-SS-MSNs) and the mesopores were functionalized with carboxylate groups for the subsequent loading of CHX. The resulting nanosystem (Ag-SS-MSNs-CHX), taking advantage of the GSH-triggered matrix degradation properties, showed sequential delivery of CHX and Ag^+^ in response to both reducing and acidic media. The efficacy of this dual-responsive nanosystem against *S**treptococcus mutans* and its biofilm was effectively demonstrated in vitro, arising values for MIC, minimum bactericidal concentration (MBC) and minimal biofilm inhibitory concentration (MBIC) of 12.5 µg/mL, 25 µg/mL and 50 µg/mL, respectively. This nanoformulation was more effective at killing *S. mutans* than an equivalent amount of free CHX. This nanoantiseptic NPs notably reduced the cytotoxicity of CHX in oral epithelial cells and did not produce abnormal effects on mice after oral exposure.

### 3.2. External Stimuli-Responsive MSNs

External stimuli, either chemical or physical, are receiving growing attention by the scientific community, since they make it possible to control antimicrobial release on demand, depending on the patient clinical needs. In this context, the external addition of certain chemical species or the application of diverse physical stimuli, such as temperature, light and alternating magnetic field (AMF) have been proposed as antimicrobial release triggers in innovative responsive nanosystems.

#### 3.2.1. Chemical Species 

It is possible to control the release of the entrapped antimicrobial cargo from MSNs on demand throughout external addition of certain chemical species. In this context, Gao and co-workers developed a dual-function multifunctional nanomaterial for bacterial detection and controllable inhibition [95]. To this end, MSNs were loaded with AMO and coated with EDA-PGEDA polymer. Then, cucurbit[7]uril (CB[7]) was attached to the resulting NPs throughout the formation of inclusion complexes with EDA moieties via ion-dipole interactions. Finally, negatively charged tetraphenylethylene tetracarboxylate (TPE-(COOH)_4_) was bond to the positively-charged supramolecular polymers on the surface of MSNs, via electrostatic interactions to for a LBL supramolecular nanoassembly. Negligible AMO release was observed from aqueous suspensions of these NPs. Nonetheless, the addition of adamantaneamine (AD) triggered AMO delivery as a consequence of the formation of a more stable supramolecular inclusion complex, AD⊂CB[7], which disrupted the nanoassembly and produced the release of EDA-PGEDA and TPE-(COOH)_4_. Moreover, aqueous suspensions of these NPs exhibited fluorescence emission, thanks to the aggregation-induced emission (AIE) effect of TPE-(COOH)_4_ onto the EDA-PGEDA layer, which allowed for the detection of bacteria. Thus, when *E. coli* and *S. aureus* were present, the competitive binding of the negatively charged bacterial surface to the positively charged EDA-PGEDA polymer broke the interactions between EDA-PGEDA and TPE-(COOH)_4_, and weakened the AIE of the latter, which led to the quenching of fluorescence emission. Moderate antibacterial activity with MIC values higher than 1000 µg/mL was observed when the nanosystem was tested against both *E. coli* and *S. aureus*. This bacterial killing capability notably increased in the presence of AD, with lower MIC values of 125 µg/mL and 250 µg/mL for *S. aureus* and *E. coli*, respectively.

#### 3.2.2. Temperature

Temperature is a physical stimulus able to that has been proposed as external release trigger for antimicrobial delivery from MSNs. Using this strategy, Martínez-Máñez and co-workers reported the design and preparation of MSNs consisting of an iron oxide (Fe_3_O_4_) core and a mesoporous silica shell, which were loaded with the antibacterial enzyme LYS and capped with poly(N-isopropylacrylamide) (PNIPAM), a thermoresponsive polymer [96]. For comparison purposes and to evaluate the efficacy of the PNIPAM capping system, loading and delivery experiments using the relative small dye tris(bipyridine) ruthenium(II) chloride (Ru(bipy)_3_^2+^) were carried out. Both nanosystems exhibited insignificant cargo release at 25 °C due to PNIPAM in hydrated extended brush form effectively closing the pores and inhibiting cargo delivery (Figure 9). However, at 37 °C, the polymer was in hydrophobic collapsed form, allowing pore uncapping and cargo delivery. Antimicrobial activity tests of LYS-loaded nanosystems at different temperatures against Gram-positive *Bacillus cereus* and *Micrococcus luteus* showed a clear temperature dependent antimicrobial activity at NPs concentration of 0.50 mg/mL. Thus, at 25 °C no toxicity was observed at a NPs concentration of 0.50 mg/mL, whereas increasing the temperature to 37 °C resulted in a reduction of bacteria growth of ca. 60% and 45% for *B. cereus* and *M. luteus*, respectively, after 24 h of assay.

#### 3.2.3. Light

The synergistic combination of phototherapy and antimicrobials delivery against bacterial infection using MSNs constitutes a promising alternative to conventional antibiotic treatments [97,98]. Following this approach, Lee and co-workers developed an antimicrobial trio-hybrid nanosystem consisting of Cu(II)-impregnated MSNs loaded with curcumin (CUR) and externally decorated with Ag NPs [97]. The resulting nanohybrid displayed excellent photodynamic inactivation (PDI) against antibiotic-resistant *E. coli* owing to the synergistic effects of silver, Cu(II) and CUR. Bactericidal effect of the nanosystem in the presence of visible light was higher than that of solely CUR or Ag NPs. Antibacterial activity enhancement was attributed to the light-controllable Ag^+^ release and the overproduction of ROS by both Ag NPs and CUR. Nanosystems with a CUR concentration of 1.5 µM provoked a ~90% eradication of the bacterial cells, accounting to the enhancement in bacterial destruction of ca. 5 log and 4 log, compared to bacterial cells treated with free CUR and Cu-MSN-AgNPs-CUR, respectively. The complete eradication of bacterial cells was accomplished by increasing the nanohybrid concentration to 20 µg/mL, which corresponded to concentration of a pure CUR of 3 µM.

In another report, Xiao and co-workers described the preparation of a multifunctional nanoplatform based on MSNs for imaging guided antimicrobial/photodynamic synergetic therapy (Figure 10) [98]. Firstly, carbon dots (C-dots) and rose bengal (RB) were embedded in core-shell MSNs affording MSN@C-dots/RB nanosystem. C-dots, acting as a fluorescence probe to attain cell fluorescence imaging, were synthetized and then embedded in MSNs, affording MSN@C-dots. Then, the RB photosensitizer, aimed to generate singlet oxygen to carry out photodynamic therapy, was loaded into MSNs. Finally, the antibiotic drug ampicillin (AMP) was loaded into MSN@C-dots/RB for synergistic bactericidal effect. Upon exposure to green light, significant chemo/photodynamic synergistic antitumor effect was reached with MSN@C-dots/RB/AMP. In addition, MSN@C-dots/RB showed remarkable bacterial inhibitory effect against *E. coli*. Thus, a concentration of MSN@C-dots/RB of 100 µg caused a delay in the bacterial growth of *E. coli* about 12 h under green light irradiation. Then, NPs were loaded with AMP and the antibacterial effect of the resulting nanosystem (MSN@C-dots/RB/AMP) on *E. coli* growth indicated that in the absence of light, an increase in the concentration of the nanosystem caused a gradual inhibition of bacterial growth, due to the antibacterial effect of AMP. However, the application of green light produced more inhibitory effect, and, thus, MSN@C-dots/RB/AMP resulted in almost complete *E. coli* eradication at concentrations of 100 µg/mL.

#### 3.2.4. Alternating Magnetic Field 

Among external stimulus, AMF is an attractive choice, since it brings the opportunity not only to control antimicrobials release from nanocarriers, but also to guide magnetic NPs to the site of infection and aid at biofilm disruption and dispersion [99,100]. Using this approach, Zink and co-workers recently designed an innovative multi-stimuli-responsive magnetic supramolecular nanoplatform for the co-delivery of large and low molecular weight antimicrobial agents to synergistically eradicate pathogenic biofilms [101]. This co-delivery nanoplatform was based on the supramolecular co-assembly of heterogeneous MSNs (Figure 11). The host MSNs (H, MSNLP@PEICD) were composed of large pore MSNs (MSNLP) capped by β-CD-modified polyethylenimine (PEICD). The guest MSNs (G, MagNP@MSNA-CB[6]) consisted of a MnFe_2_O_4_@CoFe_2_O_4_ magnetic core coated by a mesoporous silica layer decorated with both adamantine (ADA), able to interact with β-CD (on the surface of H) and N-(6-N-aminohexyl)aminomethyl triethoxysilane (AHAM), able to interact with cucurbit[6]uril (CB[6]) for efficient pore capping. The host–guest interactions between β-CD and ADA (on the surface of G) triggered the spontaneous co-assembly of H and G to form supramolecular cluster (H+G). The resulting nanoplatform was ideal to co-deliver antimicrobial agents with different molecular weight and charge, thanks to the presence of negatively charged large pores in H and positively charged small pores in G. Taking advantage of this fact, the large-molecular-weight and positively charged antimicrobial peptide melittin (MEL), and the small-molecular-weight and negatively charged antibiotic ofloxacin (OFL) were loaded in H and G, respectively. The resulting dual co-assembly (H-MEL+G-OFL) exhibited both the positively charged PEICD cap, which provided the nanoplatform of bacterium–binding and –responsive antimicrobial delivery capabilities, and the AHAM-binding CB[6] cap, which conferred the nanosystem of heat/AMF-responsive drug release ability. The excellent dual drug release efficiency of the developed nanosystem under the stimuli of pathogenic cells and AMF/heating was confirmed in *P. aeruginosa* cultures. Moreover, synergistic eradication of pathogenic biofilms by drug-loaded H-MEL+G+OFL co-assemblies under the AMF stimulus was observed. Thus, whereas H-MEL, G-OFL, or MEL+OFL reduced biofilm mass around 30–69% and killed 33–78% of pathogen cells, H-MEL+G-OFL co-assembly produced >97% biofilm mass removal and 100% pathogen cells killing. On the other hand, in vitro biocompatibility evaluation in mammalian cell cultures demonstrated that, whereas the co-assemblies exhibited excellent antibiofilm ability, they had no impact on the viability of normal cells. Finally, in vivo assays using a mouse model with implantation of biofilms-colonized silicone tubes showed that the drug-loaded co-assembly had a strong capability to eradicate pathogenic biofilms from implants, efficiently preventing infection and inflammation of host tissue. 

## 4. Conclusions

The rise of AMR and the prevalence of bacterial biofilms make bacterial infection a serious concern threatening global human health. Recent advances in nanotechnology and science of nanomaterials has allowed for engineering NPs for the treatment of infectious diseases. MSNs, particularly, exhibit enormous potential to treat bacterial infection owing to its versatility and capability to integrate multiple functionalities into one single nanoplatform. This review has tried to give a comprehensive overview of the current advances in the design and development of MSNs-based anti-infective nanotherapeutics with bacterial infection-targeting and stimuli-responsive delivery capabilities. Hence, the first part of this revision has focused on the two main developed approaches aimed at increasing antimicrobial activity and reducing off-target toxicity, i.e., targeting bacteria in planktonic or “swimming-free” state, or targeting bacterial biofilm. The first approach relies on the external functionalization of MSNs with either positively charged species to favor internalization by bacteria, or specific ligands that selectively bind bacterial surface receptors. This is of foremost interest in intracellular infections treatment, such as tuberculosis or tularemia, which requires the nanotherapeutic to distinguish between pathogenic bacteria and mammalian host cells. The second approach is committed to deal with the enormous challenge of fighting biofilm-forming bacteria. The intrinsic characteristics of biofilm matrix provide additional therapeutic opportunities and challenges to tackle infectious diseases with MSNPs. The co-delivery of substances that trigger biofilm dispersal or the grafting of biofilm-penetrating agents are two of the proposed alternatives for killing embedded pathogens by increasing their exposure to released antibiotics. The second part of this review has centered in systematically tackling the different chemical strategies aimed at incorporating pore blockers or gatekeepers that prevent premature cargo leakage and that allow its release in response to stimuli that can be either internal or external. Although triggering drug delivery using external stimuli has the advantage of providing higher control over drug dosage, using internal release triggers is less invasive. Each one has pros and cons and, therefore, they should be carefully designed and optimized, depending on clinical need. 

This review provides evidence that research on targeted stimuli-responsive MSNs against bacterial infections has experienced burgeoning growth during the last few years. The huge amount of acquired knowledge and skills on nanotechnology derived from the basic research on smart MSNs-based nanosystems for therapy and diagnosis of cancer, of course adapted to the particular characteristics of infection, has facilitated this progress. However, there are still many unsolved matters, such as the need for more specific targeting strategies for enhanced antimicrobial action, highlighting those aimed at destroying biofilms. In this sense, in vitro and in vivo studies must be carefully designed by choosing the most appropriate pathogen depending on the type of infection to be treated. Moreover, the development of better in vitro models mimicking the complex in vivo situation, such as the adsorption of a protein corona on the surface of NPs in biological milieu, is mandatory. Finally, in vivo studies require using suitable animal species for translation in human infection diseases. We are just at the beginning of an exciting scientific race, and much research effort has to be devoted to overcome the hurdles in the path from bench to bedside. 

## Figures and Tables

**Figure 1 ijms-21-08605-f001:**
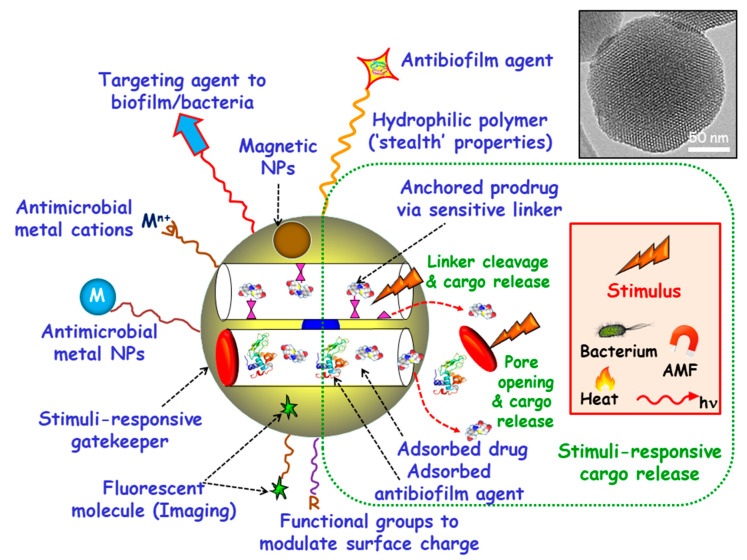
Multifunctionality of mesoporous silica nanoparticles (MSNs) for bacterial infection treatment. Targeting agents to bacteria and/or biofilm can be grafted to the outermost surface. Antimicrobial drugs and/or antibiofilm agents can be either adsorbed or grafted onto MSNs. Stimuli-responsive gatekeepers can be placed as blocking nanocaps to prevent cargo leakage. The exposition to internal (bacteria, pH, redox potential, etc.) or external (heat, light, alternating magnetic fields (AMF), etc.) stimuli triggers pore uncapping and cargo release. Antimicrobial metal NPs (M) and ions (M^n+^) can be embedded into the mesoporous structure or decorate the outermost surface of MSNs. Biocompatible hydrophilic polymers can be grafted to the external surface to provide “stealth” properties. External functionalization with different organic groups (R) can be accomplished to modulate surface charge. Magnetic NPs and fluorescent molecules can be also incorporated.

**Figure 2 ijms-21-08605-f002:**
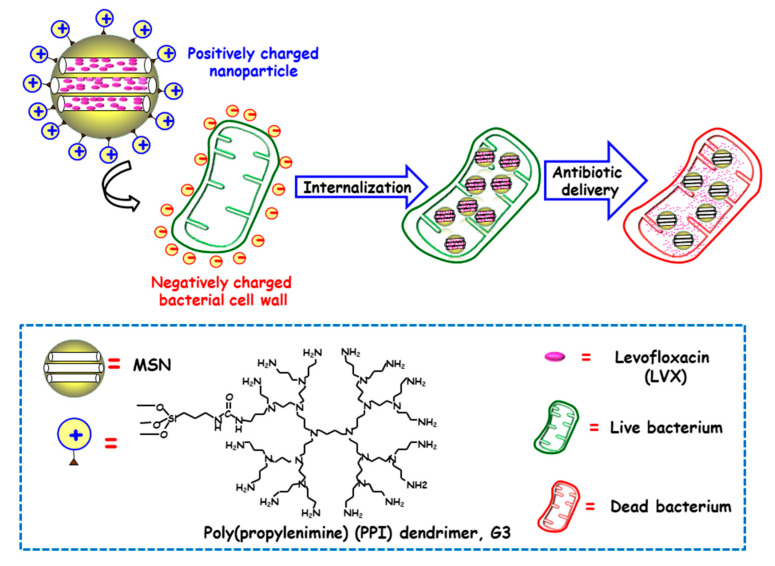
Schematic illustration of the mechanism of action of Gram negative (*E. coli*) bacteria-targeted antimicrobial nanosystems consisting of MSNs loaded with levofloxacin (LVX) and functionalized with the bacteria membrane targeting agent poly(propyleneimine) dendrimer, G3. Adapted from ref. [47]. The attractive forces between positive amine corona on MSNs and negative charges on bacterial cell wall prompt cell membrane disruption and internalization, and antibiotic delivery from MSNs produces bacteria killing.

**Figure 3 ijms-21-08605-f003:**
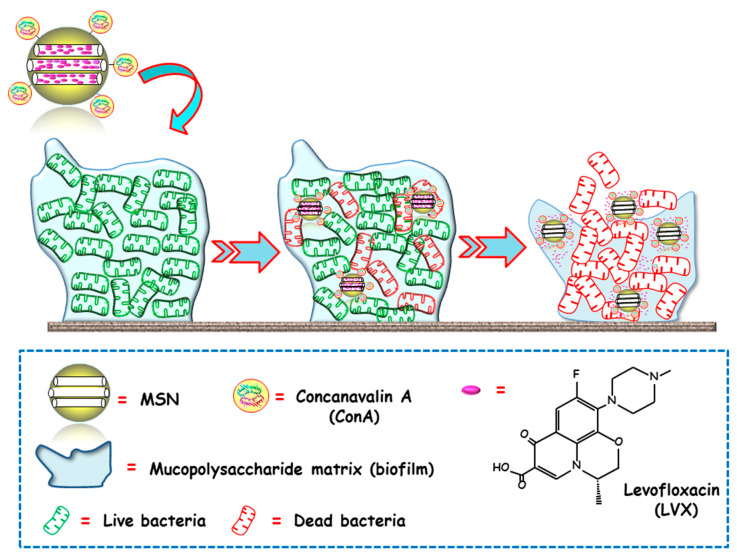
Schematic representation of the operating mechanism of bacterial biofilm-targeted antimicrobial nanosystems consisting of MSNs loaded with levofloxacin (LVX) and functionalized with the biofilm targeting agent Concanavalin A (ConA). Adapted from ref. [69]. Con A, with affinity towards certain components of the mucopolysaccharide matrix in biofilm, drives MSNs penetration into the biofilm and allows the release of the antibiotics inside to achieve antimicrobial effect.

**Figure 4 ijms-21-08605-f004:**
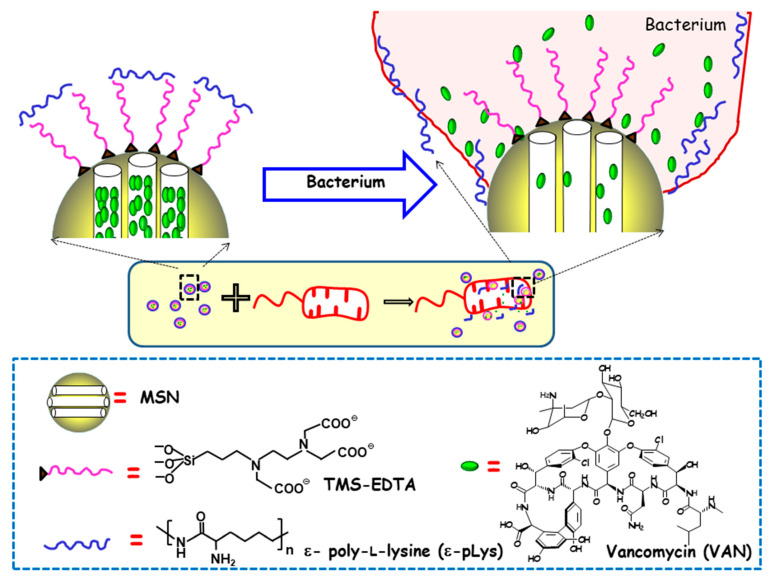
Schematic depiction of the action mechanism of bacterium-triggered antimicrobial nanosystems was composed of MSNs loaded with vancomycin (VAN), functionalized with N-[(3-trimethoxysilyl)propyl] ethylendiamine triacetic acid trisodium salt (TMS-EDTA) and capped with ε-poly-L-lysine (ε-pLys) in the presence of *E. coli.* Adapted from ref. [48]. Mesopores capping occurs via electrostatic interactions between cationic ε-pLys and negatively charged TMS-EDTA. In the presence of bacteria, the adhesion of the ε -pLys gatekeeper with the negatively charged bacteria wall triggers pore uncapping and allows cargo release, as schematically shown as an inset.

**Figure 5 ijms-21-08605-f005:**
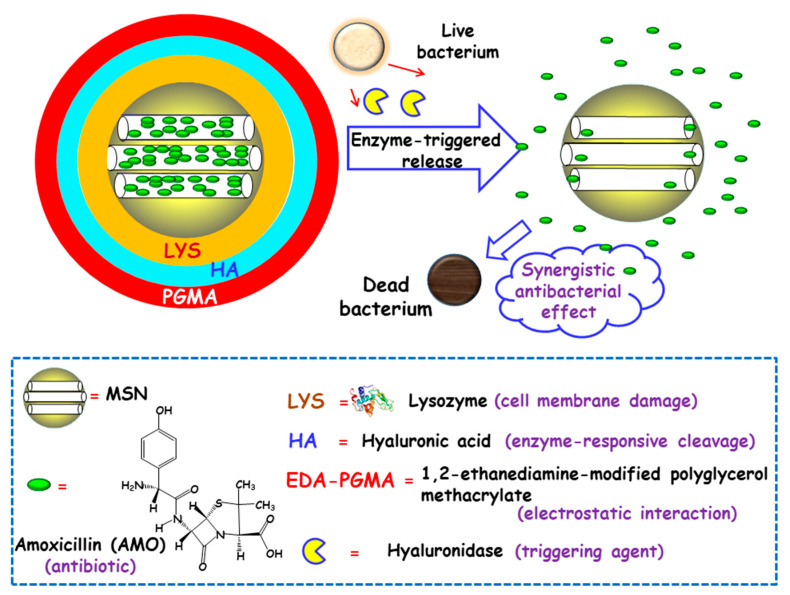
Schematic representation of the operating mechanism of enzyme-responsive antimicrobial nanosystems consisting of layer-by-layer (LBL)coated MSNs loaded with amoxicillin (AMO). Adapted from ref. [78]. Lysozyme (LYS) was electrostatically adsorbed onto AMO-loaded carboxylate-modified MSNs and then, sequential coating with hyaluronic acid (HA) and 1,2-ethanediamine-modified polyglycerol methacrylate (EDA-PGMA) was carried out. The presence of hyaluronidase enzyme secreted by bacteria triggers cleavage of the capping layer and allows the release of LYS and AMO, with synergistic antibacterial effect.

**Figure 6 ijms-21-08605-f006:**
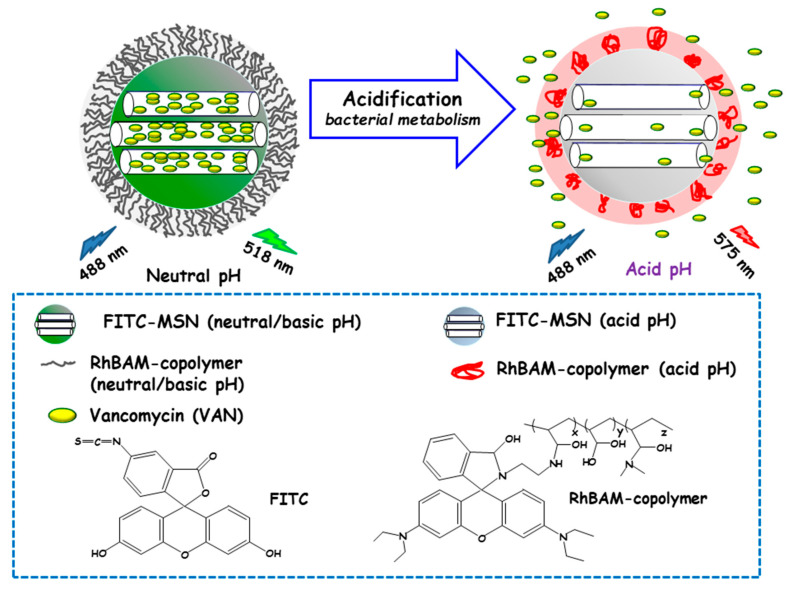
Schematic representation of the operating mechanism of pH-responsive antimicrobial nanosystems consisting of fluorescein isothiocyanate (FITC)-derivatized MSNs loaded with vancomycin and capped with a pH-sensitive hydrogel (RhBAM-modified PNIPAAm copolymer) for the efficient killing of *E. coli*. Adapted from ref. [84]. FITC: fluorescein-isotiocianate, RhBAM: Rhodamine B-derivative, PNIPAAm: poly (N-isopropyl acrylamide-co-acrylic acid). At the physiological pH (neutral) the copolymer on MSNs is in swollen-state, whereas it shrinks under environment acidification due to bacterial metabolism, allowing cargo release. The nanosystem operates as a radiometric fluorescence probe. Upon irradiation at 488 nm it shows a strong emission at 518 nm at neutral pH, whereas at acidic pH the fluorescence emission occurs at 575 nm.

**Figure 7 ijms-21-08605-f007:**
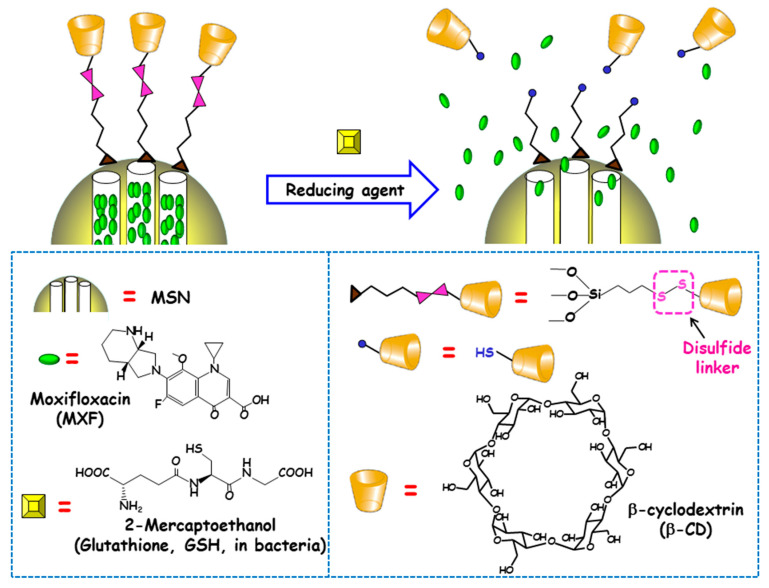
Schematic depiction of the operating mechanism of redox-responsive antimicrobial nanosystems consisting of disulfide modified MSNs loaded with moxifloxacin (MXF) and end-capped with β-cyclodextrin (β-CD) throughout a disulfide linker. Adapted from ref. [91]. The exposition to reducing milieu in bacteria (e.g. glutathione or 2-mercaptoethanol) triggers the cleavage of disulfide bond, which allows pore uncapping and cargo release.

**Figure 8 ijms-21-08605-f008:**
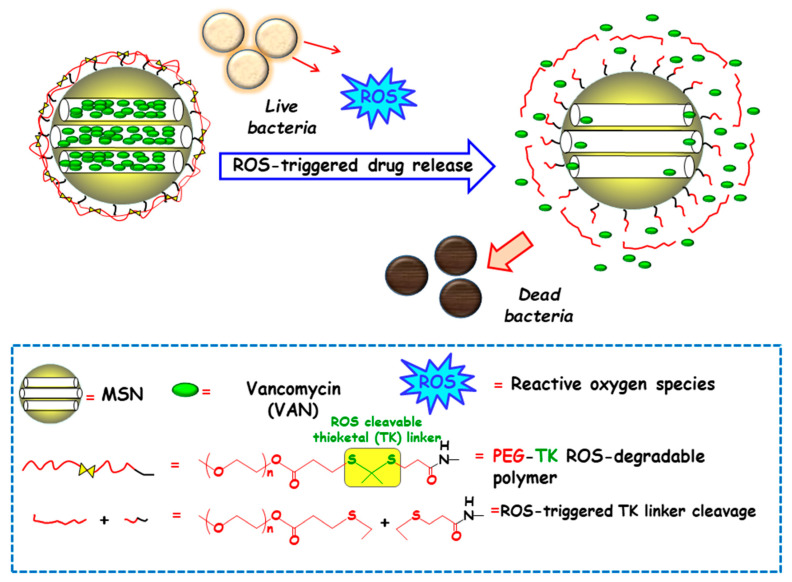
Schematic depiction of the operating mechanism of reactive oxygen species (ROS)-responsive antimicrobial nanosystems consisting of MSN loaded with vancomycin (VAN) and capped with a ROS-degradable thioketal grafted methoxy poly(ethylene glycol) (mPEG-TK) gate-like shell. Adapted from ref. [91]. The presence of ROS in the bacteria microenvironment triggers the cleavage of TK linker and the polymeric cap degradation, which produces pore uncapping, VAN release and bacteria killing.

**Figure 9 ijms-21-08605-f009:**
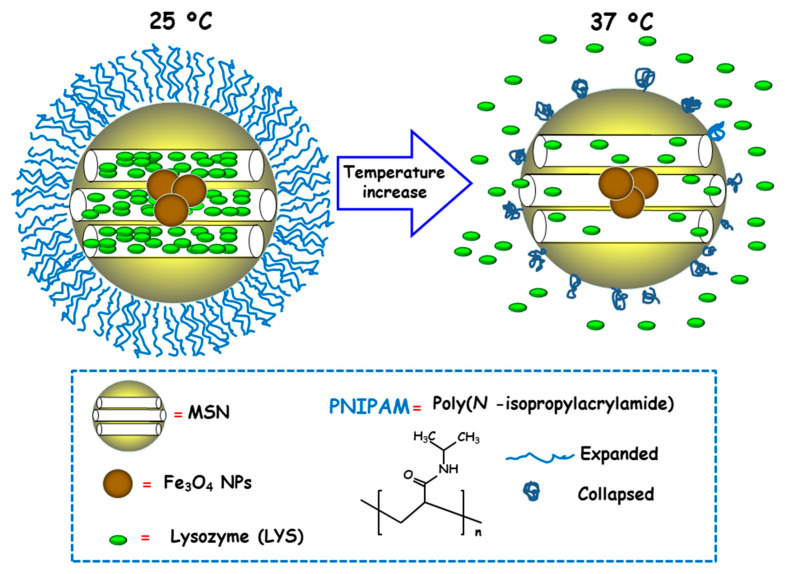
Schematic representation of the operating mechanism of temperature-responsive antimicrobial nanosystems consisting of core-shell nanoparticles (NPs) comprised of Fe_3_O_4_ cores and mesoporous silica shells, loaded with lysozyme (LYS) and coated with a thermosensitive poly(N-isopropylacrylamide) (PNIPAM) layer. Adapted from ref. [96]. At 25 ºC PNIPAM is in hydrated extended brush form, closing the mesopores and hindering cargo release. The temperature increase to 37 ºC makes the polymer to adopt a hydrophobic collapsed form, producing pore uncapping and LYS release.

**Figure 10 ijms-21-08605-f010:**
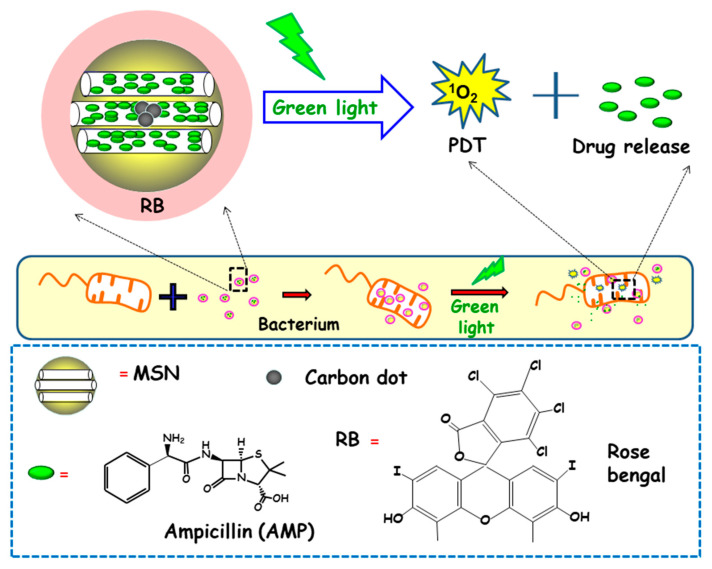
Schematic depiction of the mechanism of action of antimicrobial light-responsive antimicrobial nanosystems consisting MSNs containing carbon dots (C-dots), rose bengal (RB) and ampicillin (AMP). Adapted from ref. [98]. Upon green light irradiation of nanosystems, the RB photosensitizer triggers the generation of singlet oxygen species (^1^O_2_), which produces pore uncapping and AMP release. Synergistic combination of photodynamic therapy (PDT) and antimicrobial release lead to enhanced antimicrobial effect.

**Figure 11 ijms-21-08605-f011:**
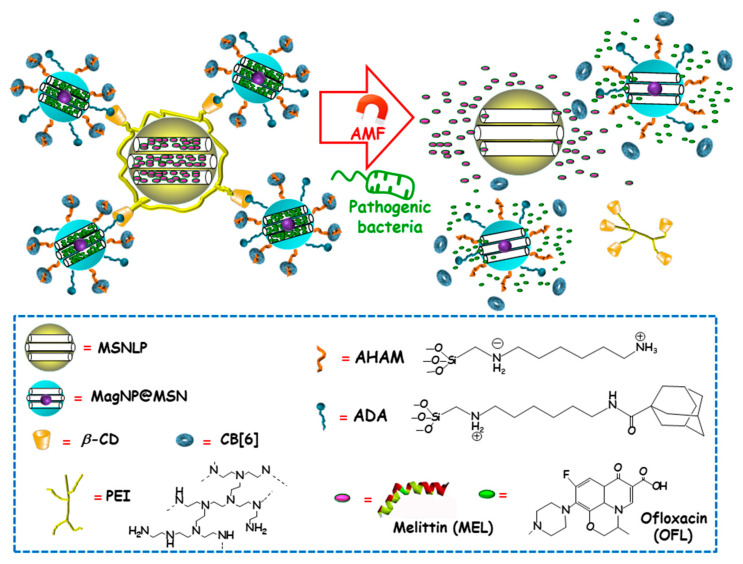
Schematic representation of the operating mechanism of dual drug-delivery and pathogen/heating/alternating magnetic field (AMF)-responsive antimicrobial nanosystems formed by the supramolecular co-assembly of host MSNs (H), guest MSNs (G). H consisted of large-pore MSNs (MSNLP) loaded with melittin (MEL) and capped by β-cyclodextrin (β-CD)-modified polyethylenimine (PEI). G consisted of MnFe_2_O_4_@CoFe_2_O_4_ magnetic NPs coated with a mesoporous silica layer (MagNP@MSN), loaded with ofloxacin (OFL) and decorated with both adamantine (ADA) (to interact with β-CD on the surface of H) and N-(6-N-aminohexyl)aminomethyl triethoxysilane (AHAM) (to interact with cucurbit[6]uril (CB[6])) for efficient pore capping. Adapted from ref. [101]. Dual antimicrobial drug release is triggered by the presence of pathogenic cells and the application of an alternating magnetic field (AMF).

**Table 1 ijms-21-08605-t001:** Bacteria-targeted MSNs as antimicrobial delivery systems.

Targeting Ligand ^1^	Drug Loaded ^2^	Nanocarrier ^3^	Bacteria ^4^	Assay	Ref.
G3	Levofloxacin	MCM-41 G3-MSNs	*E. coli*	In vitro	[47]
ε-pLys	Vancomycin	MCM-41 ε-pLys-MSNs	*E. coli*	In vitro	[48]
ε-pLys	HKAIs	MCM-41 ε-pLys-MSNs	*E. coli,* *S. marcescens*	In vitro	[49]
FB11	Model drugs (Fluorescein, Hoechst 33342)	MCM-41 FB11mFt LPS-MSNs	*F. tularensis*	In vitro	[50]
Anti-*S. aureus* Ab	Vancomycin	Ab@S-HA@MMSNs	*S. aureus*	In vitro	[51]
SA20hp	Vancomycin	MCM-41 SA20hp-MSNs	*S. aureus*	In vitro	[52]
UBI_29–41_	Gentamicin	MSN-LU	*S. aureus*	In vitro & in vivo	[53]
LL-37	Colistin	MSN@LL-(LL-37)	*P. aeruginosa*	In vitro	[54]

Trehalose	Isoniazid	M-PFPA-Tre	*M. smegmatis*	In vitro	[55]
Trehalose	Isoniazid	Tre-HOMSNs	*M. smegmatis*	In vitro	[41]
Arginine	Ciprofloxacin	Arg-MSNs	*S. typhimuruim*	In vitro & in vivo	[56]
Folic acid	Ampicillin	MSN@FA@CaP@FA	*E. coli*, *S. aureus*	In vitro & in vivo	[57]
Vancomycin	Vancomycin (grafted)	MCM-41 MSNs⊂VAN	*S. aureus*	In vitro	[58]

^1^ G3: poly(propyleneimine) third-generation dendrimer; ε-pLys: ε-poly-L-lysine cationic polymer; FB11: FB11 antibody for lipopolysaccharide (LPS) present in *Francisella tularensis (Ft)*; Anti-*S. aureus*: *S. aureus* antibody; SA20hp: SA20 aptamer with hairpin structure; UBI_29–41_: Ubiquicin; LL-37 peptide: Human cathelicidin peptide; Arg: arginine. ^2^ HKAIs: histidine kinase authophosphorylation inhibitors. ^3^ MCM-41 G3-MSNs: MCM-41 type MSNs functionalized with G3; MCM-41 ε-pLys-MSNs: MCM-41 type MSNs functionalized with pLys; MCM-41 FB11mFt LPS-MSNs: MCM-41 type MSNs functionalized with FB11 antibody through a derivative of the O-antigen of Ft LPS; Ab@S-HA@MMSNs: Sulfonated-hyaluronic acid (S-HA) terminated magnetic MSNs modified with Anti-*S. aureus* (Ab); MCM-41 SA20hp-MSNs: MCM-41 type MSNs functionalized with SA20hp; MSN-LU: MSNs modified with a lipidic bilayer surface shell and conjugated with UBI_29–41_; MSN@LL-(LL-37): MCM-41 type MSNs coated with a lipidic layer and conjugated with LL-37; M-PFPA-Tre: Perfluorophenylazide-functionalized decorated with α,α-trehalose; Tre-HOMSNs: Trehalose-functionalized hollow oblate mesoporous silica nanoparticles; Arg-MSN: MCM-41 type MSNs functionalized with *L*-Arg; MSN@FA@CaP@FA: MSNs covered by double folic acid (FA) and calcium phosphate (CaP) layers; MCM-41 MSNs⊂VAN: MCM-41 type MSNs functionalized with vancomycin. ^4^
*E. coli*: *Escherichia coli; S. marcescens: Serratia marcescens; F. tularensis: Francisella tularensis; S. aureus: Staphylococcus aureus;*
*P. aeruginosa: Pseudomonas aeruginosa; M. smegmatis: Mycobacterium smegmatis; S. typhimuruim*: *Salmonella typhimuruim*.

**Table 2 ijms-21-08605-t002:** Bacterial biofilm-targeted MSNs as antimicrobial delivery systems.

Targeting Ligand ^1^	Drug Loaded	Nanocarrier ^2^	Bacterial Biofilm ^3^	Assay	Ref.
DAMO	Levofloxacin	MCM-41 DAMO-MSNs	*S. aureus*	In vitro	[47,68]
G3	Levofloxacin	MCM-41 G3-MSNs	*E. coli*	In vitro	[47]
ConA	Levofloxacin	MCM-41 ConA-MSNs	*E. coli*	In vitro	[69]

^1^ DAMO: N-(2-aminoethyl)-3-aminopropyltrimethoxy-silane; G3: poly(propyleneimine) third-generation dendrimer; ConA: concanavalin A. ^2^ MCM-41 DAMO-MSNs: MCM-41 type MSNs functionalized with DAMO; MSNs-ConA: MCM-41 type MSNs decorated with ConA. ^3^
*E. coli: Escherichia coli; S. aureus: Staphylococcus aureus.*

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
