# Peer review of "Targeted Stimuli-Responsive Mesoporous Silica Nanoparticles for Bacterial Infection Treatment"

_ijms, 2020, doi:10.3390/ijms21228605_

Round 1

Reviewer 1 Report

The manuscript by Montserrat Colilla and María Vallet-Regí is a very very complete review addressing useful topic - Mesoporous Silica Nanoparticles as a drug delivery agent. The most important references were considered. The works summarizes the actual status of the technology describing the strategies evaluated in recent years with relevant work covering most of the work done. The subject fits the journal scope and, in my opinion, is interesting for the scientific community. The introduction part brings the reader perfectly into the field. Conclusions are clear and schematic.

In my opinion the MS deserves publication.

Author Response

Thank you. We sincerely acknowledge the referee’s comments and we are pleased to receive such a good consideration of our work.

Reviewer 2 Report

the review is really well organized and interesting, schemes are clear and fully understandable. the references reviewd are quite updated, and the view on the field seems almost complete.

a couple of minor points for the introductory part:

in the introduction, when talking about inorganic NP and stimuli responsive antibacterial action, authors should cite at least a couple of examples about the use of noble metal nanoparticles  photothermal treatment for bacteria, for example see RSC advances 6 (74), 70414-70423

when talking about decoration os silica surfaces nwith antibacterial metal cations, a review has appeard which should be cited, see  Coordination Chemistry Reviews 275, 37-53

Author Response

Response to Reviewer 2 Comments

The review is really well organized and interesting, schemes are clear and fully understandable. the references reviewd are quite updated, and the view on the field seems almost complete.

We really acknowledge the referee for his/her carefully reading and we are glad to hear such positive comments.

a couple of minor points for the introductory part:

in the introduction, when talking about inorganic NP and stimuli responsive antibacterial action, authors should cite at least a couple of examples about the use of noble metal nanoparticles  photothermal treatment for bacteria, for example see RSC advances 6 (74), 70414-70423

We appreciate the referee comments aimed at improving the overall information provided by the review. We have incorporated 2 new references (now references 37 and 38), as suggested by the referee, and a short text to explain them, as follows (page 3, lines 81-84):

Antimicrobial metal NPs can be either embedded in the mesoporous structure or decorating the outermost surface of MSNs. In the case of metal noble NPs (mainly from gold and silver) their well-known antimicrobial action can be reinforced, when needed, by the antibacterial action based on the photo-thermal effect upon proper laser excitation (D'Agostino et al., RSC Adv. 2016, 6, 70414-70423; Xu et al., Nanoscale 2019, 11, 8680-8691)

when talking about decoration os silica surfaces nwith antibacterial metal cations, a review has appeard which should be cited, see  Coordination Chemistry Reviews 275, 37-53

We acknowledge the referee for providing us such reference. We have included it in the text, and slightly modified a sentence in the manuscript (page 3, lines 84-85), as follows:

“Antimicrobial metal cations can be complexed to ligands grafted to the surface of MSNs (Pallavicini et al. Coord. Chem. Rev. 2014, 275, 37-53,)”